# HDAC6 Inhibition Extinguishes Autophagy in Cancer: Recent Insights

**DOI:** 10.3390/cancers13246280

**Published:** 2021-12-14

**Authors:** Eugenia Passaro, Chiara Papulino, Ugo Chianese, Antonella Toraldo, Raffaella Congi, Nunzio Del Gaudio, Maria Maddalena Nicoletti, Rosaria Benedetti, Lucia Altucci

**Affiliations:** 1Department of Precision Medicine, University of Campania “Luigi Vanvitelli”, 80138 Naples, Italy; eugenia.passaro@unicampania.it (E.P.); chiara.papulino@unicampania.it (C.P.); ugo.chianese@unicampania.it (U.C.); antonella.toraldo@unicampania.it (A.T.); raffaella.congi@studenti.unicampania.it (R.C.); nunzio.delgaudio@unicampania.it (N.D.G.); mariamaddalena.nicoletti@studenti.unicampania.it (M.M.N.); 2Biogem Institute of Molecular and Genetic Biology, 83031 Ariano Irpino, Italy

**Keywords:** autophagy, epigenetics, HDAC inhibitors, HDAC6, drug resistance, cancer

## Abstract

**Simple Summary:**

Autophagy is an essential process in cell recycling, and its involvement in cancer has been increasingly recognized in the last few decades. This mechanism acts as a double-edged sword in tumor progression and is known to either block or promote tumorigenesis in a context-specific manner. Its role in determining chemotherapeutic resistance makes it a potential target in cancer treatment. The two autophagic inhibitors hydroxychloroquine and chloroquine are currently used in the clinic but cause several side effects in tumor patients. Since recent studies also show that epigenetic enzymes such as histone deacetylase (HDAC) proteins are able to modulate autophagy, this review focuses on the ability of HDAC6 to actively regulate the autophagic process. We also explore the possibility of using HDAC6 inhibitors as therapeutic agents in adjuvant treatment or in combination with autophagic modulators to trigger this mechanism, thus avoiding the occurrence and effects of chemoresistance.

**Abstract:**

Autophagy is an essential intracellular catabolic mechanism involved in the degradation and recycling of damaged organelles regulating cellular homeostasis and energy metabolism. Its activation enhances cellular tolerance to various stresses and is known to be involved in drug resistance. In cancer, autophagy has a dual role in either promoting or blocking tumorigenesis, and recent studies indicate that epigenetic regulation is involved in its mechanism of action in this context. Specifically, the ubiquitin-binding histone deacetylase (HDAC) enzyme HDAC6 is known to be an important player in modulating autophagy. Epigenetic modulators, such as HDAC inhibitors, mediate this process in different ways and are already undergoing clinical trials. In this review, we describe current knowledge on the role of epigenetic modifications, particularly HDAC-mediated modifications, in controlling autophagy in cancer. We focus on the controversy surrounding their ability to promote or block tumor progression and explore the impact of HDAC6 inhibitors on autophagy modulation in cancer. In light of the fact that targeted drug therapy for cancer patients is attracting ever increasing interest within the research community and in society at large, we discuss the possibility of using HDAC6 inhibitors as adjuvants and/or in combination with conventional treatments to overcome autophagy-related mechanisms of resistance.

## 1. Introduction

Autophagy is a cellular homeostasis mechanism of “self-digestion” triggered in response to physiological changes such as starvation, growth factor signaling, or cellular stress [1]. The autophagic process is responsible for the degradation and elimination of unnecessary, misfolded, or non-functional components in the cytoplasmic environment, recycling them to maintain a biological balance [2]. In nutritional deficient state, the autophagic process is crucial for the proper functioning of cellular activities [3]. The basic functions of autophagy are associated with the regulation of energy metabolism, and its role in recycling macromolecules is related to its ability to degrade different substrates, thus allowing the process to potentially activate mainly the central carbon metabolism [4,5]. In this context, glycolysis can be promoted via the degradation of carbohydrates into sugar, or the tricarboxylic acid cycle (TCA) can be activated by obtaining proteins from amino acids. The autophagy mechanism provides metabolic substrates, for instance, by obtaining amino acids via the degradation of proteins or lipids from liquid droplets, thereby contributing to mitochondrial metabolism [6].

Autophagy can be divided into three major types: macro-autophagy (also known as basal autophagy), micro-autophagy and chaperone-mediated autophagy (CMA), all of which promote the degradation of cytosolic components at the lysosome but with distinct pathways [7]. In macro-autophagy (henceforth referred to as ‘autophagy’), a portion of cytoplasm—including organelles—is enclosed by a phagophore (isolation membrane) to form an autophagosome, a double-membraned structure whose outer membrane subsequently fuses with lysosomes to become autolysosomes, in which the internal material is degraded [8]. In contrast, micro-autophagy is the direct engulfment of lysosomes through invagination of the lysosomal membrane [9]. In CMA, recyclable cargo is delivered in a complex with chaperone proteins—recognized by lysosomal-associated membrane protein 2A (LAMP-2A) receptor—across the lysosomal membrane [10]. Since autophagy is essential to maintaining cell metabolism and energy homeostasis, it is not surprising that it has an important role in human health and disease.

## 2. Autophagy and Cancer

### 2.1. Autophagy and Tumorigenesis

Autophagy supports biological processes in physiological conditions, while its activity has a negative effect in numerous diseases, including cancer [11,12]. Many forms of cancer are characterized by a marked autophagic activity [13] that modulates the recycling of proteins and thus promotes uncontrolled cell proliferation. In this way, the breakdown of cellular components promotes cell survival by maintaining cellular energy levels in starved and non-starved cancer cells [14]. Cells use autophagy to prevent tumorigenesis but also, in some cases, to promote tumor cell survival. The role of autophagy in cancer is dynamic and controversial and is also partly time/context dependent [15]. In the first stages of carcinogenesis, autophagy acts as a suppressive mechanism [16,17,18]. It can inhibit cancer progression by blocking tumor initiation [19] and by promoting survival and the growth of tumor cells in late stages [20]. Increased autophagy is also associated with the promotion of cancer cell death and the suppression of tumor cells. Metabolic changes associated with autophagy enable cancer cells to survive, making them highly dependent on this process.

Autophagy has an important function in tumor metabolism. Cancer cells displayed an increase in the levels of autophagy necessary to meet the metabolic requirements enabling tumorigenesis in vitro and in vivo [21,22]. In nutrient deprivation conditions, the catabolic capacity of autophagy supports cell viability [23]. Increased autophagic activity was also observed in tumor regions poor in nutrients compared to widely vascularized areas [24]. In addition, the ability of cells to recycle damaged components through autophagy is a defense mechanism against chemotherapeutic drugs or radiotherapy, allowing cancer cells to resist drug-related damage, thereby promoting survival and tumor growth [25] and inducing drug resistance [26]. Autophagy is therefore part of a removing strategy to overcome stresses induced by anticancer agents and is involved in resistance to cancer treatments [27]. The degradation process by autophagy can be broadly defined as either non-selective or selective. In the former case, upon nutritional demands and stress conditions, cells eliminate components without distinguishing them in the cargo in order to meet energy requirements [28]. This mechanism is considered the canonical autophagy process, where Unc-51 like autophagy activating kinase 1 (ULK1) complex is required for autophagy-related (ATG) protein recruitment at the phagophore assembly site [29]. Unlike non-selective autophagy, the selective defense process based on the tagging and targeting of cargos with ubiquitination is well known [30]. Although selective cargoes are mainly damaged proteins, other waste materials are recognized and tagged via a ubiquitination mechanism including mitochondria (mitophagy), peroxisomes (pexophagy), lipid droplets (lipophagy), glycogens (glycophagy), ribosomes (ribophagy), endoplasmic reticulum (ER-phagy) and pathogens such as bacteria, fungi, parasites and viruses (xenophagy). Ubiquitinated contents are then bound by ubiquitin-binding protein adaptors such as sequestosome 1 (SQSTM1/p62) and delivered to the assembling autophagosomes [31].

### 2.2. Autophagy and Cancer Treatment Resistance

Conventional chemo/radiotherapy can be combined with autophagy modulators [6]. In xenograft models and several cancer cell lines, treatment with different chemical agents was able to prevent the upregulation of autophagy—as a survival mechanism—by synergizing with autophagy inhibitors and exerting an anticancer effect, strongly blocking tumor growth [32]. Many therapeutic agents stimulate autophagy, favoring cancer metabolism and preventing DNA damage, thus allowing the promotion of tumor drug resistance [26]. Since the controversial role of autophagy both in promoting tumor growth and supporting tumorigenic processes and in protecting against carcinogenesis is well recognized, it is reasonable to speculate that this pathway could represent a valid therapeutic target. The effect of the autophagic process in terms of its cytotoxicity or cytoprotective function is mostly influenced by the cellular context [33]. In cancer cells, autophagy is considered a double-edged sword, playing pro-death and/or pro-survival roles [26]. Determining cellular context is therefore crucial to the specific choice of anticancer therapy based on the modulation of autophagy mechanisms. The regulation of autophagy involves p53-related pathways, mitogen-activated protein kinase (MAPK)-related pathways, metabolic stress-induced signaling, microRNA-triggered signaling and long noncoding RNA-triggered signaling, thus making autophagy a highly unpredictable mechanism in terms of therapy responsiveness [26,34].

### 2.3. Therapeutic Agents Targeting Autophagy

Many clinical and preclinical studies have highlighted the efficacy of autophagic inhibition as a support treatment in cancer. Chloroquine and hydroxychloroquine target the autophagic pathway and have already been clinically used for cancer therapy in combination treatment, despite their reported side effects [35]. Specifically, although chloroquine is described as increasing chemotherapy efficacy, its systemic effect could also sensitize kidney cells, leading to acute kidney injury [36]. These findings indicate that new approaches are required and have prompted the research community to further explore the field and look for alternative strategies. Autophagy is a key process in cancer progression, and multiple studies have highlighted the involvement of epigenetic mechanisms in its regulation. Epigenetic mechanisms such as DNA methylation, histone modifications and long noncoding RNA presence are able to modulate autophagy in a dynamic interplay. Although autophagy is primarily a cytoplasmic event, a recent study reported that its transcriptional and epigenetic regulation occurs in the nucleus and is critical in the process [37]. The identification of a transcriptional factor mainly involved in autophagic mechanisms, the ubiquitin-binding histone deacetylase (HDAC) 6, led to the hypothesis of intervening epigenetically. In this review, we discuss the role of epigenetic HDAC-mediated modifications in controlling autophagy in cancer, focusing on the controversy surrounding its ability to promote or to block tumor progression. We explore how this enzyme is involved in the recycling mechanism regulating cancer cell proliferation in the context of autophagy. We also discuss the possibility of using HDAC6 inhibitors as adjuvants in combination with conventional treatments to overcome autophagy-related mechanisms of resistance.

## 3. HDACi Trigger the Autophagy Process

The autophagic process is known to be mediated by HDAC inhibitors (HDACi) [38]. HDAC proteins differ in their structure, mechanism of action, substrate selectivity, cellular localization and tissue-specific expression. The HDAC family comprises 18 enzymes that are grouped into four different classes based on sequence homology to yeast. Class I HDACs include HDAC1, 2, 3 and 8; these proteins are mainly located in the nucleus and are involved in cell proliferation and survival. Members of the class II HDAC family have tissue-specific roles depending on phosphorylation status. These enzymes can shuttle between the cytosol and nucleus and are further divided into two subgroups: class IIa, comprising HDAC4, 5, 7 and 9, which can shuttle between the cytoplasm and nucleus, and class IIb, including HDAC6 and 10, predominantly located in the cytoplasm. HDAC6 has also been identified in the nucleus. Only HDAC6 is specific for α-tubulin, an important protein required for cell mitosis and movement. Class III HDACs, known as sirtuins 1–7, require nicotinamide adenine dinucleotide as a coenzyme for their reaction. Class IV HDACs, exhibiting features of class I and II, include only HDAC11, mainly localized in the nucleus, and are involved in regulating interleukin-10 expression [39]. These enzymes play a pivotal role in the reversible control of histone and non-histone protein acetylation, thus contributing to the block of cancer initiation and progression. The aberrant expression of class I, II and IV HDACs is associated with a variety of cell transformation, including solid and hematological malignancies. High levels of HDACs were observed in advanced stages of tumor disease and are also associated with poor patient outcome and prognosis [40].

HDACi are therefore considered promising anticancer drugs, especially since they can be used in combination with chemotherapeutics and/or radiotherapy [41]. These compounds are in general small molecules that do not inhibit all HDAC isoforms in the same manner and are classified as either pan- or “isotype-specific” compounds, referred to as class I-specific inhibitors [42]. Pan-HDACi targeting classes I, II and IV are hydroxamic acids (TSA, SAHA, LAQ824, CBHA) and pyroxamic acids (PXD101, CRA-026440) effective against HDACs in the nanomolar range. In contrast, class I-specific HDACi are carboxylic acids (valproic acid, sodium butyrate), benzamides (MS275, CI-994, MGCD0103) or cyclic tetrapeptides (trapoxin, depsipeptide, spiruchostatin A) [43]. A list of HDACi in clinical trials adapted from Rikiishi [42] and Chen et al. [44] is reported in Table 1. HDACi are currently used in the clinic as anticancer agents to alter the acetylation of both histone and non-histone targets [45]. However, their use is still limited due to their non-specific and non-selective mode of action. Recent findings also showed the ability of HDACi to mediate the induction of both apoptosis [46] and autophagy [47] associated with anticancer activity in a variety of cancer cell lines. Epigenetic modifications and autophagy are in fact regarded as two interlinked cellular processes that play important roles in cancer development [48]. Although several HDACi are known to induce autophagy [38,49], their therapeutic effects are still debated, and the underlying mechanisms of action need to be clarified.

In vitro studies found that SAHA (vorinostat) induced autophagy in hepatocellular carcinoma cells, blocking proliferation, and in glioblastoma cells, inducing apoptotic and non-apoptotic cell death [38,49,50]. SAHA treatment enhanced autophagy via the suppression of the mammalian target of rapamycin (mTOR) complex 1, a key component involved in regulating the autophagic process, coordinating the balance between autophagy and cell growth [51]. mTOR blocks autophagy through phosphorylation and consequently the inactivation of the ULK1 complex, an upstream component of autophagy pathway [52]. Thus, through mTOR inactivation, SAHA reestablishes ULK1 function and consequently leads to autophagy induction. However, the mechanism underlying mTOR inactivation by SAHA remains largely unknown. On one side, mTOR suppression by SAHA may occur via modulation of the acetylation of essential non-histone proteins; on the other side, it could be directly imputable to upregulation of transcription of microtubule-associated protein 1A/1B-light chain 3 (*LC3*), one of the essential autophagy genes [50]. SAHA was also shown to increase expression of the autophagic component LC3 and to inhibit mTOR in glioblastoma cells [38]. Furthermore, autophagy-mediated cell death was found to be induced by (i) the conversion of unconjugated LC3 (LC3-I) to conjugated LC3 (LC3-II), (ii) the transfer of LC3 to autophagosomes, (iii) the increase in acidic vesicular organelles and (iv) the expression of autophagy-related proteins and the autophagy-related *Atg5* gene [53,54,55].

Treatment with HDACi was found to cause caspase-independent autophagic cell death in different cancer models [1]. This effect was observed in HeLa cells treated with SAHA and sodium butyrate. Although the underlying mechanism is unknown, caspase-3 activation and cell death induced by HDACi were revealed by ultrastructural changes in cells [56]. These two compounds also induced apoptosis when caspase activation was abolished, showing that HDACi-induced caspase-independent cell death is related to autophagic cell death. These findings confirm that HDACi are able to cause both caspase-dependent apoptosis and caspase-independent autophagic cell death. Treatment with HDACi can trigger the autophagy process and is thus able to affect apoptosis resistance. In the early phases of tumorigenesis, autophagy could suppress cell transformation, inhibiting cancer inflammation and restricting tumor necrosis [57].

Another critical mechanism associated with HDACi-mediated autophagy is reactive oxygen species (ROS) accumulation. ROS production seems to be an important mechanism for autophagy activation [58]. High levels of ROS produce an increase of metabolism and—under metabolic stress conditions—autophagy is consequently induced to maintain cellular integrity [59]. ROS regulates autophagy through the upregulation of Beclin-1, the oxidation of ATG4 and mitochondrial dysfunction [60,61,62]. ROS generation was also found in combination with mTOR attenuation; SAHA treatment interrupted mitochondrial respiration and energy metabolism, leading to massive ROS accumulation and autophagy induction [63]. The proteasome inhibitor bortezomib and the HDACi romidepsin are reported to synergistically provoke apoptotic cell death, inducing autophagic cytotoxicity [64]. Although the link between ROS formation and HDAC inhibition is unclear, it is known to be related to post-translationally modified regulatory proteins [65]. Other studies corroborate this hypothesis; the balance between HDAC and histone acetyltransferase (HAT) enzymes was found to modulate the autophagy process [66,67]. The p300-mediated acetylation of various components of autophagy machinery was found to play an inhibitory role in this context [68]. Many ATG proteins were in fact shown to be regulated by acetylation due to specific HAT–HDAC equilibrium. In nutrient deprivation conditions—for example, in the absence of growth factors—glycogen synthase kinase-3 activates the HAT KAT5 (TIP60), which directly acetylates and stimulates ULK1 [69]. In contrast, treatment with HDACi in Down syndrome-associated myeloid leukemia repressed autophagy, probably due to the downregulation of *ATG7* gene, was associated with its hyperacetylation [70]. Specifically, mono treatment with valproic acid, SAHA, TSA, panobinostat or JQ2, a specific HDAC1/2 inhibitor, increased acetylation levels of ATG7, promoted a decrease in autophagic flux, and induced apoptosis in myeloid leukemia cells [70].

Several selective HDACi targeting autophagy are already being investigated in clinical trials (NCT01266057, NCT02316340). Since it is recognized that HDACi are able to impact the autophagic process in different ways and that this process is specifically regulated by HDAC6 [71].

As previously mentioned, the acetylation process is important in the regulation of autophagy, and the main “actor” modulating this process through interaction with microtubule proteins is HDAC6. What is special about HDAC6 is that it contains two functional catalytic domains, and both domains are homologous and functionally independent of the overall activity of HDAC6 [72]. Additionally, the C-terminal end of HDAC6 contains a ubiquitin-binding zinc finger domain (ZnF-UBP domain, also known as the PAZ, BUZ or DAUP domain) that is related to the regulation of ubiquitination-mediated degradation. Furthermore, differently from other HDACs, HDAC6 is mainly localized in the cytoplasm because of the presence of a nuclear export sequence [72]. Given the role of HDAC6 in the autophagic processes described above, therapies based on HDAC6 inhibition may represent a potential approach to block cancer cell survival by triggering autophagic pathways.

## 4. Role of HDAC6 in Autophagic Processes

HDAC6 is tightly involved in autophagy in many conditions. This enzyme is an important player in macro-autophagy and regulates crucial events associated with chloroquine-mediated autophagy inhibition [73]. Several findings support the idea that epigenetic marks are able to modulate autophagy and therefore that their dysregulation can subsequently lead to an inappropriate autophagic process. This deregulation is therefore thought to underlie the etiopathogenesis of human diseases such as neurological disorders (Alzheimer’s and Parkinson’s disease), neurodegenerative and age-related conditions and various cancer types [37,74,75].

Among all HDAC enzymes, HDAC6 is increasingly emerging as a pivotal player in autophagy since it has been associated with the modulation of different pathways related to this biological process (Figure 1) [76]. HDAC6 works in cooperation with the ubiquitin–proteasome system (UPS), favoring the processing of misfolded proteins [77]. HDAC6 is the only HDAC family member with intrinsic ubiquitin-binding activity in addition to its deacetylase action [78]. It is known to deacetylate α-tubulin and increase cell motility through its deacetylase activity on non-histone substrates (Figure 2a) [79]. HDAC6 binds cortactin and promotes the polymerization of F-actin, influencing microtubule-dependent cell motility [80]. The impact of HDAC6 on autophagy has been correlated with several biological steps, making it a key regulator. The first mechanism involves the modulation of heat shock protein 90 (HSP90) activity, as this chaperone is a substrate for HDAC6 deacetylase activity (Figure 2b) [81]. In vitro experiments showed that the shuttling of polyubiquitinated substrates is modulated by HDAC6, which coordinates autophagosome activity in aggresome formation [77]. Additionally, HDAC6 might favor the transport of lysosomes to the site of autophagy, considering that its knockdown results in the dispersal of lysosomes differently from the microtubule organizing center [82].

As previously mentioned, different types of autophagy mechanisms are described in mammalian cells, in all of which HDAC6 has a role. Macro-autophagy, also known as basal autophagy, is the most studied autophagic system. This process involves the sequestration of substrates in cytosolic double-membrane vesicles called autophagosomes [8]. Superfluous and damaged proteins, cytosolic proteins and invasive microbes are then degraded in the autophagosomes, and breakdown products are released into the cytosol in order to recycle macromolecular constituents and generate energy for maintaining cell viability [83,84]. The unique morphological feature of macro-autophagy is that autophagosomes form de novo by expansion and not from a pre-existing organelle already containing cargo [85]. The formation of autophagosomes in mammalian cells starts upon the induction of the autophagic process at multiple sites throughout the cytoplasm rather than at a single phagophore assembly site, as in yeast [86]. HDAC6 is a key component in macro-autophagy, targeting protein aggregates and misfolded organelles [73]. Due to its ubiquitin-binding ZnF-UBP domain present at the C-terminal end, HDAC6 also plays an important role in ubiquitination-mediated degradation [87]. HDAC6 is therefore considered a regulator of two systems responsible for the degradation of misfolded and damaged proteins [82]. This mechanism is regulated via UPS and autophagic machinery. HDAC6 acts at multiple levels, including the ultimate clearance of aggresomes, corroborating the involvement of HDAC6 in autophagy [82,88]. Ubiquitinated damaged proteins are recognized by HDAC6, which modulates autophagosome–lysosome formation and the elimination of autophagic substrates through cortactin-actin proteins [89]. Indeed, HDAC6 deacetylase activity contributes to the production of the deacetylated form of cortactin, which interacts with F-actin to form cortactin–F-actin; this complex is then recruited to the MTOC, promoting autophagosome–lysosome formation and degradation [80]. The HDAC6–actin interaction is primarily linked to the regulation of cell motility [80,90]. Its association with the microtubule network and ubiquitinated proteins led to the surprising finding that HDAC6 is a regulatory component of aggresomes, the MTOC-localized inclusion body where excess protein aggregates are deposited [77]. Aggresomes containing misfolded ubiquitinated proteins co-localize with the MTOC in which the autophagosome and lysosome fuse [91]. Aggresome formation is associated with the protection of cells, promoting the elimination of toxic aggregates where the HDAC6 BUZ finger domain facilitates the transport of cargo by the microtubule network [77,92].

p62 has a controversial role in autophagy. If on one side it is involved in the removal of misfolded proteins by inhibiting HDAC6 activity, on the other it regulates the elimination of the cortactin–F-actin assembly in the MTOC [93]. Findings demonstrated that in the absence of p62, the complex cortactin–F-actin is confined in the periphery and there is an accumulation of ubiquitinated proteins [94]. Downregulation of HDAC6 inhibits the fusion between the autophagosome and lysosome and subsequent protein aggregation [73]. Protein assembly and transport are regulated by p62, whereas HDAC6 modulates autophagolysosomes.

The homeostasis of autophagy is maintained by the p62/HDAC6 ratio. While protein-mediated toxic stress is enforced by proteasome inhibition, cell fate in response to an altered ratio of p62 to HDAC6 remains unclear.

Another autophagy-related pathway is the CMA system, which differs in its selectivity due to the KFERQ motif, a pentapeptide that allows for the identification of its target from among the plethora of possible cytoplasmic targets. Indeed, target proteins containing the KFERQ-like motif are recognized by a maxi-protein complex that orchestrates all the steps, from the recognition phase up to fusion with the lysosome, which guarantees its degradation. Heat shock cognate 70 (HSC70) is responsible for identifying the KFERQ-like motif and is assisted in the cargo and lysosomal anchoring phase by a structure of chaperones cooperating in the CMA process. Interestingly, among the chaperones and other molecules that exert this autophagic activity, several have been recognized as substrates of HDAC6. The acetylation state of HSC70 modulates binding with other important proteins that trigger the autophagic process. Its acetylation is known to reduce the expression of Beclin-1, an important promoter of phagophore formation, inhibiting the assembly of the ATG12–ATG5 complex, and of LC3, thus reprogramming autophagy in an inhibited state. HSC70 is overexpressed in many tumors [95], and its ability to modulate the stability of numerous proteins alters cellular control and apoptosis mechanisms, inducing tumor development. These findings clearly indicate that the epigenetic contribution of HDACs can have an effect in this context. HDAC6 was shown to deacetylate HSC70 and its co-chaperone DNAJA1, influencing their interaction [96]. HSP90, a well-known HDAC6 target, is responsible for coordinating the cargo phase of misfolded proteins, and HDAC6 inhibition results in an HSP90 hyperacetylation state, leading to a loss of chaperone activity [97,98]. Evidence suggests that HDAC6 influences the activity of CMA by deacetylating HSP90, leading to the dissociation of the macro-complex. Blocking HDAC6 deacetylase activity affects the signaling cascade of successive targets of HSP90 such as LAMP2, a pivotal protein involved in the autophagosome–lysosome fusion step, indirectly favoring the remodulation of the autophagic protein profile [99].

Another autophagic process involving HDAC6 is mitophagy. Mitophagy is a cargo-specific autophagy process in which mitochondria are removed [100]. In macro-autophagy, once these components are sealed in autophagosomes fused with lysosomes, they are eliminated, and the degradation of these contents in lysosomes provides elements to be recycled to produce ATP [101]. This mechanism plays a pivotal role in the reestablishment of cellular homeostasis in both physiological and stress conditions. Specifically, it has a protective function since damaged mitochondria can stimulate innate immunity, favoring an increase in ROS species or the release of mitochondrial DNA [102]. This process is modulated by two genes, *Parkin* and PTEN-induced kinase 1 (*PINK1*). In normal conditions, PINK1 is transported to mitochondria and degraded by mitochondrial proteases. In the presence of defective mitochondria, PINK1 is not degraded and is stabilized on the outer membrane. In response, Parkin translocates to the site of damaged mitochondria [103,104] and then ubiquitinates outer membrane mitochondrial proteins, causing their fragmentation. The fragments are identified by autophagosome and removed following lysosomal fusion [104,105,106]. Alternatively, p62 and HDAC6 recruit damaged mitochondria to the isolation membrane by interacting with LC3 [100]. HDAC6 is reported to be involved in the binding of ubiquitinated substrates, and it accumulates in the mitochondria, as it is required for Parkin-mediated autophagy [101].

Since HDAC6 also plays an important role in mitophagy and HDAC inhibitors are able to directly target ATG proteins interfering with this pathway, it is reasonable to speculate that HDAC6 inhibition affects mitophagy, leading to a block of tumor proliferation and self-renewal [70].

## 5. Discussion

HDAC6 is an unconventional enzyme compared to other HDACs since it is the only isozyme that contains two catalytic domains, CD1 and CD2 [107]. It is localized in the cytoplasm, as indicated by a serine/glutamate-rich repeat, and through the CD2 domain catalyzes the deacetylation of K40 on α-tubulin in the lumen of the microtubule [108,109]. HDAC6 regulates the deacetylation of various substrates observed in many diseases, such as cancer and neurodegenerative disorders. HDAC6 interacts with numerous cytoplasmatic partners (α-tubulin, cortactin, and HSP90) regulating cytoskeleton-dependent and chaperone-dependent cellular functions related to the aggresome–autophagy pathway, and, conversely, these interacting partners influence HDAC6 activity [110]. HDAC6 is also reported to regulate tau proteins found in tauopathy-associated neurodegenerative diseases. In neuropathologies, these factors are usually hyperphosphorylated, and HDAC6 knockdown drives a decrease in tau phosphorylation and accumulation [110].

Given the importance of HDAC6 in regulating autophagic-related pathways, specifically in the microtubule trafficking of aggresome substrates and autophagic components, the idea that it may represent an important therapeutic target triggering the autophagy process is gaining ground [110]. HDAC6 is involved in tumor cell survival and growth, prompting the development of many selective inhibitors. However, since the in vivo and in vitro results using these agents as monotherapies were not particularly promising due to their selectivity and side effects, they are being investigated for use in combinatory treatments.

Several autophagy-related compounds have demonstrated efficacy but are still in the early stage of development. These compounds were found to either stimulate or inhibit different targets of the autophagy process at several levels of disease. At the very early stage of tumor transformation, autophagy plays a cytoprotective role, blocking inflammation, necrosis and tumorigenesis [111,112]. By contrast, in late-stage cancers, autophagy prevents tumor cell survival by inhibiting protein recycling and cell growth [112]. Discovering new agents is therefore very challenging, since the effect of autophagy processes is highly complicated and context specific. Recent findings highlight that antitumor agents, and in particular autophagy inhibitors, can be used in combinatory treatments to enhance the overall effect on cell death [113,114,115,116]. Numerous studies in fact report that resistance to anticancer treatment can be overcome by inhibiting autophagy via RNA interference and/or by using autophagy inhibitors. Most preclinical studies demonstrated the ability of antitumor agents to trigger protective autophagy when used in combination with autophagy inhibitors [116,117]. These agents have different origins and are classified as natural compound, synthetic and tyrosine kinase inhibitors.

Currently, several small molecules are under development to increase their target specificity against HDAC6. A recent preclinical study described the design of a novel HDAC6-selective inhibitor, C1A, based on the structure of the naturally occurring hydroxamate trichostatin A and SAHA [118]. C1A was shown to have an antiproliferative effect in human colon cancer HCT116 cells. In this study, the effect of the compound on substrates of HDAC6 involved in the autophagy process such as α-tubulin and HSP90 led to its activation. The cellular impact of C1A was also investigated in HCT116 cells to elucidate the mechanism of growth inhibition. Successive studies evaluated the effect of C1A on autophagy and cell death [119]. Analysis of autophagic markers showed increased expression of LC3-II, LC3-I and p62 in HCT116 and osteosarcoma cells treated with C1A, demonstrating its capability to induce apoptosis via the inhibition of the autophagic flux [119]. Another recently reported highly selective HDAC6 inhibitor is J22352, shown to exert anticancer effects in glioblastoma [120]. J22352-mediated HDAC6 downregulation induced a reduction in cell migration, enhancing autophagic cell death. The main action observed was HDAC6 degradation through the accumulation of p62, promoting the delivery of ubiquitinated HDAC6 for proteasomal degradation. These findings reveal that J22352 significantly induces the accumulation of autophagic vacuoles and consequently inhibits autophagy in glioblastoma cells. This molecule acts by preventing the deacetylation of α-tubulin in the cytoplasm and thus inhibits cancer metastasis. Autophagic cancer cell death was the result of increased metabolic stress due to the autophagosome–lysosome fusion blockade. However, very few HDAC6 modulators are currently being investigated in clinical trials. In a phase I trial, a potent and selective small molecule inhibitor of HDAC6, KA2507, displayed both cell intrinsic and extrinsic anticancer activity (NCT03008018). The trial is currently focusing on the pharmacokinetics, safety and tolerability of this molecule used in monotherapy in advanced cancer patients (melanoma and colorectal cancer) [121]. Very few HDAC6 inhibitors are reported to trigger autophagy, but not all HDAC6 inhibitors modulate only this process [119]. ACY-1215 (ricolinostat), for instance, is a small selective HDAC6 inhibitor with minimal clinical activity as a single agent. This compound is considered a first-in-class isoform-selective HDAC6 inhibitor and is being studied for the treatment of patients with lymphoma [122]. Based on preclinical studies, ricolinostat mediates its action through the unfolded protein response (UPR) system. Several studies describe its effect in combination with paclitaxel for the dual targeting of microtubules. In addition, a very recent clinical study reported dose-limiting toxicities for ricolinostat, with side effects mainly associated with gastrointestinal symptoms [123]. In a multi-center phase Ib clinical trial, ricolinostat was administered in combination with lenalidomide and dexamethasone to 38 relapsed/non-responder patients with multiple myeloma, demonstrating an improvement in this context [72]. Given the key epigenetic involvement of HDAC6 in the autophagic process and the controversial role of autophagic machinery in promoting or blocking tumor development, designing a tailored therapy for different cancer types based on the use of specific HDAC6 inhibitors could be a valid approach. HDAC6 inhibitor therapy should ideally be used in the early stages of cancer promotion to block cancer proliferation and development by triggering autophagy leading to cancer cell death. In 2017, Li et al. reported the ability of a combination therapy using tubastatin A, a selective HDAC6 inhibitor, and temozolomide (TMZ), a “cornerstone” compound for the treatment of glioblastoma multiforme, to overcome multidrug resistance issues associated with endoplasmic reticulum stress tolerance in glioma cells. These two agents acted synergistically, determining an increase in pro-apoptotic signals from the UPR, and modulated ubiquitination–autophagy turnover involved in the degradation and clearance of ubiquitinated misfolded proteins [111]. Furthermore, under chemotherapy stress conditions, cells activate autophagic defence mechanisms, making them more resistant to the effects of anticancer treatments.

## 6. Conclusions

In this review, we discuss autophagy processes and the implications related to the role of HDAC6. Since it is clear that recycling activity can benefit a system with a high rate of proliferation such as cancer, it is equally reasonable that HDAC6 works as the main actor in this signaling. Although autophagic function appears to be associated with particular tumor types, the involvement of HDAC6 in the autophagy pathway leaves no doubt as to its role as a possible target in cancer-specific models. The use of selective drugs against HDAC6 in combination and/or as adjuvant therapy could therefore represent an effective strategy to enhance tumor cell chemosensitivity by interfering with the autophagic process, thus promoting cell death.

## Figures and Tables

**Figure 1 cancers-13-06280-f001:**
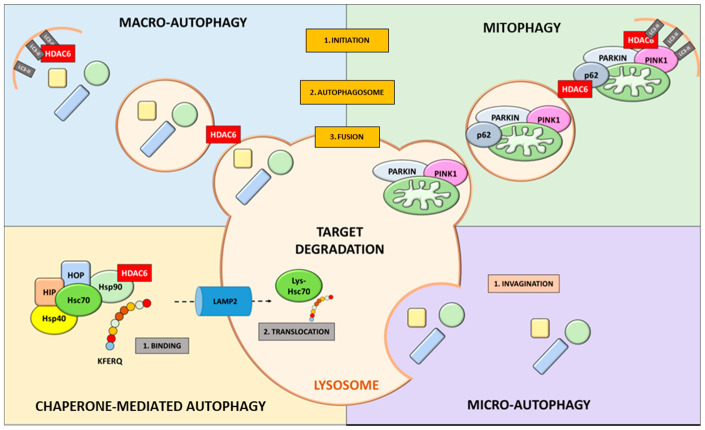
HDAC6 involvement in autophagy-related mechanisms.

**Figure 2 cancers-13-06280-f002:**
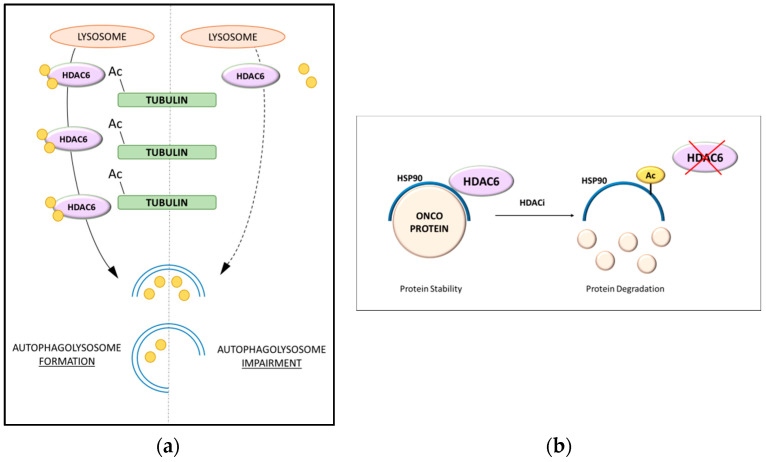
HDAC6 inhibition in autophagy pathways: (**a**) impairment in autophagolysosome assembly upon α-tubulin deacetylation and (**b**) acetylated HSP90 leads to the degradation of oncoproteins.

**Table 1 cancers-13-06280-t001:** Properties of HDAC inhibitors in clinical trials.

Compound	Target	Source	Chemical Class	Isoform Selectivity	Study Phase - Clinical trial
**Vorinostat (SAHA)** **Belinostat (PXD-101)** **Panobinostat (LBH-589)** **Trichostatin A (TSA)** **Quisinostat (JNJ-16241199)** **WW437** **Dacinostat (LAQ824)**	Pan-HDAC	Synthetic Synthetic Synthetic Natural Synthetic Synthetic Synthetic	Hydroxamic acid	Class I, II and IV Class I and II Class I, II and IV Class I and II Class I and II HDAC 2 and 4 Class I and II	FDA approval for Cutaneous T-Cell Lymphoma FDA approval for Peripheral T Cell lymphoma (PTCL) FDA approval (PTCL and multiple myelomas) Toxic–Phase I Phase II Cutaneous T-Cell Lymphoma Pre-clinical Phase I
**Pivaloyloxmethylbutyrate (AN-9)** **Sodium Butyrate (NaB)** **Sodium Phenylbutyrate** **(4-PB)** **Valproate (valproic acid)**	Pan-HDAC	Synthetic Natural Synthetic Synthetic	Short chain fatty acids	Class I and IIa Class I and IIa Class I and IIa Class I and IIa	Phase II Melanoma and Phase I Chronic lymphocytic leukemia (CLL) Phase I Colorectal cancer FDA approval (urea cycle disorders) Phase I (Brain and Central Nervous System Tumors)
**Romidepsin (FK228)**	Pan-HDAC	Natural	Cyclic tetrapeptides	Class I (HDAC1, 2, 4, 6)	Phase II
**Entinostat (MS-275)** **Tacedinaline (CI-994)** **Mocetinostat (MG-0103)**	Pan-HDAC	Synthetic Synthetic Synthetic	Benzamides	Class I Class I Class I and IV	Phase II (Hodgkin’s Lymphoma) Phase II (Myeloma) Phase I (Hodgkin’s Lymphoma)
**Trapoxins (TPX)** **α-ketoamides** **Heterocyclic ketones**	Pan-HDAC	Natural Synthetic Synthetic	Ketones	Class I NA NA	NA (Not approved) NA NA
**Cambinol** **EX-527** **Sirtinol** **Nicotinamide**	Sirtuin inhibitors	Synthetic Synthetic Synthetic Synthetic		SIRT1 and 2 SIRT1 and 2 SIRT1 and 2 class III	Pre-clinical Pre-clinical Pre-clinical Phase III (laryngeal cancer)
**Azelaic Bishydroxamic Acid (ABHA)** **m-carboxycinnamic acid bis-hydroxamide (CBHA)** **Ricolinosta (ACY-1215)** **Tubacin**	Selective HDAC	Synthetic Synthetic Synthetic Synthetic	Hydroxamate Derivatives Benzamides	HDAC3 HDAC3 HDAC6 HDAC6	NA Pre-clinical Phase II Pre-clinical Acute lymphocytic leukemia (ALL)

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
