# Peer review of "HDAC6 Inhibition Extinguishes Autophagy in Cancer: Recent Insights"

_cancers, 2021, doi:10.3390/cancers13246280_

Round 1
Reviewer 1 Report
This review paper by Passaro et al., reviews literature on possible inhibition of cancer progression by HDAC6 inhibitors via inhibition of underlying autophagy mechanism. Furthermore, the authors explore the plausible role of HDAC6 inhibitors as adjuvant and/or in combination with conventional anti-cancer therapeutics. An additional figure showing interaction/participation of HDAC6, HDAC6 inhibitor, autophagy in cancer pathway can be included to represent the working hypothesis of the paper.
Author Response
We would like to thank the reviewers for their thoughtful review of our manuscript. They raise important issues, and their input has been very helpful in improving the quality of the manuscript. We agree with almost all of their comments and have revised the manuscript accordingly. The hypothesis and implications of our work are now stated more clearly. We have followed all the reviewers’ suggestions and clarified the text when necessary. We feel confident that the new version of the manuscript is greatly improved. Below is a detailed point-by-point response to each of the reviewers’ comments, indicating how/where we have revised the text.
REVIEWER 1 COMMENTS:
- This review paper by Passaro et al., reviews literature on possible inhibition of cancer progression by HDAC6 inhibitors via inhibition of underlying autophagy mechanism. Furthermore, the authors explore the plausible role of HDAC6 inhibitors as adjuvant and/or in combination with conventional anti-cancer therapeutics. An additional figure showing interaction/participation of HDAC6, HDAC6 inhibitor, autophagy in cancer pathway can be included to represent the working hypothesis of the paper.
- We thank the reviewer for their useful suggestion. We have now added an additional figure (Figure 2, line 384) to illustrate our hypothesis about the interaction/participation of HDAC6, HDAC6 inhibitors, and autophagy in cancer pathway.
Reviewer 2 Report
In this paper Passaro and co-authors wrote a review exploring the possibility of using HDAC6 inhibitors in combination with autophagy modulators to prevent chemo-resistance. The review is full of information, but it seems that the authors made many statements and conclusions that are not very well supported.
These are few issues that need to be addressed:
- The authors stated that Saha inhibited apoptosis referring to the reference 54. This reference does not show any data on Saha inhibiting apoptosis. Strong data was published showing that Saha indues apoptosis in many cancers. The authors maybe want to state that Saha induce apoptotic and non-apoptotic cell death. This needs to be addressed and the conclusion changed.
- In the beginning of the discussion section, the authors state that unlike other HDACs, HDAC6 plays a role in epigenetic modifications of both histone and non-histone proteins,… These effects are not unique to HDAC6. This needs to be corrected.
- Some vocabulary and spellings mistakes need to be fixed through the text. For example, in lines 156, 310, and 330.
Author Response
We would like to thank the reviewers for their thoughtful review of our manuscript. They raise important issues, and their input has been very helpful in improving the quality of the manuscript. We agree with almost all of their comments and have revised the manuscript accordingly. The hypothesis and implications of our work are now stated more clearly. We have followed all the reviewers’ suggestions and clarified the text when necessary. We feel confident that the new version of the manuscript is greatly improved. Below is a detailed point-by-point response to each of the reviewers’ comments, indicating how/where we have revised the text.
REVIEWER 2 COMMENTS:
1. The authors stated that SAHA inhibited apoptosis referring to the reference 54. This reference does not show any data on SAHA inhibiting apoptosis. Strong data was published showing that SAHA indues apoptosis in many cancers. The authors maybe want to state that SAHA induce apoptotic and non-apoptotic cell death. This needs to be addressed and the conclusion changed.
- We are grateful to the reviewer for highlighting this important point. We have now corrected the text accordingly in line 192. We have also added an additional reference (PMID: 22493260) which reports that autophagy inhibition can sensitize cells to both apoptotic and non-apoptotic cell death induced by SAHA.
2. In the beginning of the discussion section, the authors state that unlike other HDACs, HDAC6 plays a role in epigenetic modifications of both histone and non-histone proteins… These effects are not unique to HDAC6. This needs to be corrected.
- We agree with the reviewer. We have now substituted the sentences from line 388 to 391 with: “HDAC6 is an unconventional enzyme compared to other HDACs. Since it is the only isozyme which contains two catalytic domains, CD1 and CD2 (PMID: 27454933). It is localized in the cytoplasm as indicated by a serine/glutamate-rich repeat (PMID: 15347674) and through the CD2 domain catalyzes the deacetylation of K40 on α-tubulin in the lumen of the microtubule (PMID: 27454933)”.
3. Some vocabulary and spellings mistakes need to be fixed through the text. For example, in lines 156, 310, and 330.
- We have carefully revised the manuscript for typos and for vocabulary and spelling mistakes, including, as suggested, in lines 156, 310, and 330.

Reviewer 3 Report
This review article focuses on an important topic with significant clinical relevance. However, it is poorly organized. Major revision is necessary before it can be considered for publication.
For a review article, a well-organized, clear structure is very important to allow the readers to comprehend current state of knowledge about the reviewed topic. This review has 4 sections, including the introduction and discussion sections. Adding some sub-titles and clearly defined sub-topics may significantly improve the clarity of this review article. The content needs to be rearranged under clearly defined sub-titles.
The "Introduction" section is too long and not organized with clear structure, therefore, this section lacks clarity. This may be improved by having a shorter summary as the introduction section and a separate section about autophagy and cancer. The introduction section can focus on autophgy in general. The part of autophagy and cancer also needs to be better organized. For example, it could been arranged under the subtitles like "autophagy and tumorigenesis", "autophagy and cancer treatment resistance", and "therapeutic agents targeting autophagy".
The author should first introduce the role of HDACs in autophagy with the focus on HDAC6 in cancer-related autophagy, and then introduce the inhibitors (HSACi) with focus on their mechanisms of action and therapeutic potential.
Line 138-154, the HDAC6 and HDACi paragraph can be put in the next section.
Line 27, define "epi-drugs".
Author Response
We would like to thank the reviewers for their thoughtful review of our manuscript. They raise important issues, and their input has been very helpful in improving the quality of the manuscript. We agree with almost all of their comments and have revised the manuscript accordingly. The hypothesis and implications of our work are now stated more clearly. We have followed all the reviewers’ suggestions and clarified the text when necessary. We feel confident that the new version of the manuscript is greatly improved. Below is a detailed point-by-point response to each of the reviewers’ comments, indicating how/where we have revised the text.
REVIEWER 3 COMMENTS
1. This review article focuses on an important topic with significant clinical relevance. However, it is poorly organized. Major revision is necessary before it can be considered for publication. For a review article, a well-organized, clear structure is very important to allow the readers to comprehend current state of knowledge about the reviewed topic. This review has 4 sections, including the introduction and discussion sections. Adding some sub-titles and clearly defined sub-topics may significantly improve the clarity of this review article. The content needs to be rearranged under clearly defined sub-titles. The "Introduction" section is too long and not organized with clear structure; therefore, this section lacks clarity. This may be improved by having a shorter summary as the introduction section and a separate section about autophagy and cancer. The introduction section can focus on autophagy in general. The part of autophagy and cancer also needs to be better organized. For example, it could be arranged under the subtitles like "autophagy and tumorigenesis", "autophagy and cancer treatment resistance", and "therapeutic agents targeting autophagy".
- We thank the reviewer for their invaluable comments and useful suggestions. We have modified and reorganized the main text accordingly, adding some subtitles to better illustrate the topics discussed in this review. The introduction section has now been modified to focus on the mechanism of autophagy in general. We have also reorganized section entitled “Autophagy and cancer” into different subsections: “Autophagy and tumorigenesis” “Autophagy and cancer treatment resistance”, and “Therapeutic agents targeting autophagy”.
2. The author should first introduce the role of HDACs in autophagy with the focus on HDAC6 in cancer-related autophagy, and then introduce the inhibitors (HDACi) with focus on their mechanisms of action and therapeutic potential.
- We thank the reviewer for their suggestion. However, the idea behind the manuscript is to first introduce HDACs and their inhibitors in the section “HDACi TRIGGER THE AUTOPHAGY PROCESS” and then focus on HDAC6 as the main actor in autophagy processes, leading into a discussion on the possibility of using HDAC6 modulators to overcome mechanisms of drug resistance, which is the pivotal point of the review. We believe that the review is better structured in this way.
3. Line 138-154, the HDAC6 and HDACi paragraph can be put in the next section.
- We agree with the reviewer and have modified the text accordingly: the original lines 138–144 have now been moved to the section “ROLE OF HDAC6 IN AUTOPHAGIC PROCESSES” (new lines 266–273) to better clarify the importance of this enzyme in modulating autophagy in disease.
- Furthermore, we have moved lines 149-152; 153–155 to the section “HDACi TRIGGER THE AUTOPHAGY PROCESS” (new lines 246–249) to describe the role of HDAC6 in regulating cancer development and the possibility of using a modulator with conventional treatment to overcome autophagy-related mechanisms of resistance.
4. Line 27, define "epi-drugs".
- We apologize for any lack of clarity. We have substituted “epi-drugs” “with epigenetic modulators” in the abstract (line 27).

Round 2
Reviewer 3 Report
The manuscript has been sufficiently improved to warrant publication in Cancers.